# APEX1 Expression as a Potential Diagnostic Biomarker of Clear Cell Renal Cell Carcinoma and Hepatobiliary Carcinomas

**DOI:** 10.3390/jcm8081151

**Published:** 2019-08-01

**Authors:** Ji-Myung Kim, Min-Kyung Yeo, Jae Sung Lim, In-Sang Song, Kwangsik Chun, Kyung-Hee Kim

**Affiliations:** 1Department of Laboratory Medicine, Chungnam National University Hospital, 282 Munwha-ro, Daejeon 35015, Korea; 2Department of Pathology, Chungnam National University School of Medicine, 266 Munhwa Street, Daejeon 35015, Korea; 3Department of Urology, Chungnam National University School of Medicine, 266 Munhwa Street, Daejeon 35015, Korea; 4Department of Surgery, Chungnam National University School of Medicine, 266 Munhwa Street, Daejeon 35015, Korea; 5Department of Surgery, Chungnam National University Hospital, 282 Munwha-ro, Daejeon 35015, Korea

**Keywords:** apurinic/apyrimidinic endonuclease 1/redox effector factor 1, hepatocellular carcinoma, cholangiocarcinoma, clear cell renal cell carcinoma, diagnosis

## Abstract

Apurinic/apyrimidinic endonuclease 1/redox effector factor 1 (APEX1) has been known to play key roles in DNA repair, the regulation of diverse transcriptional activity, and cellular responses to redox activity. This study aimed to examine serum APEX1 (s-APEX1) expression as a possible screening biomarker for clear cell renal cell carcinoma (ccRCC), hepatocellular carcinoma (HCC), and proximal and distal cholangiocarcinoma (CC). A total of 216 frozen serum samples were collected from 39 healthy control cases, 32 patients with ≥58 copies/mL of hepatitis B viral DNA (HBV DNA (+)), 40 ccRCC cases, 59 HCC cases, and 46 CC cases. The serum samples were examined for s-APEX1 concentration by enzyme-linked immunosorbent assay. The association of APEX1 expression with clinicopathological characteristics was also studied by immunohistochemical staining in 106 ccRCC, 131 HCC, and 32 intrahepatic CC cases. The median s-APEX1 concentrations of the HCC, CC, ccRCC, healthy control, and HBV DNA (+) groups were 0.294, 0.710, 0.474, 0.038, and 2.384 ng/mL, respectively (*p* < 0.001). Univariate and multivariate analyses revealed that increased cytoplasmic APEX1 expression led to a shorter disease-free survival period in HCC and CC cases. We suggest that the s-APEX1 level could be a potential diagnostic biomarker of ccRCC, HCC, and CC. Additionally, cytoplasmic APEX1 expression in cancer cells could be used to predict relapses in patients with HCC or CC.

## 1. Introduction

Apurinic/apyrimidinic endonuclease 1/redox effector factor 1 (APEX1) is a multifunctional protein that acts as a transcriptional coactivator in apoptosis, proliferation, and differentiation. It is also an essential factor in the DNA base excision repair pathways response to DNA damage and a negative regulator of the Ras-related GTPase Rac1 activity to control intracellular reactive oxygen species [1]. APE1 refers to a member of the DNA base excision repair pathway; Ref-1 refers to redox regulation on diverse redox-dependent mechanisms [1]. APEX1 has been associated with cancer, cardiovascular diseases, and neurodegeneration and seems to play a role in the inflammatory process. One study demonstrated that APEX1 induces interleukin-8 production by gastric cells in *Helicobacter pylori* infection [2], and another reported that APEX1 is a key upstream regulator in TLR2-dependent keratinocyte inflammatory responses [3].

APEX1 protein is ubiquitous, but altered levels have been found in a variety of cancers, such as germ cell tumors, rhabdomyosarcomas, breast carcinomas, hepatocellular carcinomas (HCCs), non-small cell lung carcinomas, ovarian serous carcinomas, gastric carcinomas, esophageal squamous cell carcinomas, and bladder carcinomas [4,5]. Although APEX1 subcellular expression is mainly nuclear, cytoplasmic localization of APEX1 has been associated with different functional processes depending on the condition [1]. The extracellular occurrence of APEX1 in the serum or plasma, which could result from tumor cells or immune cells, has been evaluated in HCC, bladder cancer, and non-small-cell lung cancer [6,7,8]. The blood is a good source of biomarkers for screening or follow-up. The kidneys and liver have significant blood flow, so the effectiveness of using a serologic biomarker to diagnose cancer in these organs may be particularly high.

We hypothesized that the expression levels of subcellular or extracellular serum APEX1 (s-APEX1) could be the basis for a diagnostic or therapeutic strategy. This study aimed to evaluate subcellular APEX1 expression in cancer cells and the extracellular APEX1 level in serum as a possible diagnostic or prognostic biomarker in patients with clear cell renal cell carcinoma (ccRCC), HCC, or cholangiocarcioma (CC), compared to the levels in nontumor groups, including a healthy control group and patients with ≥58 copies/mL of hepatitis B viral DNA (HBV DNA (+)).

## 2. Materials and Methods

### 2.1. Patients and Samples

The study considered a total of 505 samples from South Korean individuals with ccRCC, HCC, or CC (Table 1) who had not received preoperative chemotherapy or radiation therapy. All 505 tissue or serum samples were provided by the National Biobank of Korea at Chungnam National University Hospital, a member of the Korean Biobank Network. The frozen serum samples from cancer patients were obtained before operation for surgical resection.

The formalin-fixed, paraffin-embedded (FFPE) tissue samples used in this study were surgically resected between 1999 and 2014 at Chungnam National University Hospital in Daejeon, South Korea. All medical records of the patients were reviewed by Kim, K-H, and Kim, J-M to obtain clinical data. The tumor, node, and metastasis (TNM) staging and histologic grading for ccRCC, HCC, and CC were determined at the time of surgical resection and were based on the 8th edition of the American Joint Committee on Cancer (AJCC) staging system [9]. ccRCC and hepatobiliary cancer recurrence or metastasis was determined via histological findings. Disease-free survival (DFS) was determined as the time between the date of initial surgical resection and the date of the cancer’s recurrence or metastasis. Overall survival (OS) was defined as the time from the initial surgical resection until the date of death due to any cause. Without death, recurrence, or metastasis confirmation, OS or DFS time was stopped at the last known date that the patient was alive. All paired tissue samples from the HCC, CC, and ccRCC patients consisted of a tumor tissue section and a matched nontumor tissue section, at least 3.0 cm from the tumor mass. Tissue microarrays (TMAs) were constructed as previously described [10].

Quantitation of HBV DNA was performed using the Cobas HBV assay for the Cobas 6800 system (Roche Diagnostics GmbH, Mannheim, Germany). The analytical measurement range for HBV DNA levels was 58 to 5,820,000 copies/mL (10 to 1,000,000,000 IU/mL), and samples that had 58 or more HBV DNA copies/mL were defined as positive for HBV DNA (HBV DNA (+)).

This study was approved by the Institutional Review Board of Chungnam National University Hospital (CNUH 2018-06-030-002). All frozen serum or tissue samples and clinical data were obtained from the National Biobank of Korea at Chungnam National University Hospital. All patients signed a written informed consent form for biobanking before data were included in the register. The requirement for informed consent for the retrospective comparison study was waived because the study was based on immunohistochemical analysis using FFPE tissue.

### 2.2. Frozen Serum APEX1 Measurement by Enzyme-Linked Immunosorbent Assay (ELISA)

ELISA was conducted as previously described using an APEX1 enzyme-linked immunosorbent assay kit (Cat#MR-APE064-HS. MediRedox Inc., Daejeon, Korea), including the mouse monoclonal anti-human APEX1 antibody (Cat#LS-C392589, LifeSpan BioSciences, WA, USA) and the rabbit polyclonal anti-human APEX1 antibody (MediRedox, Daejeon, Korea) [11].

### 2.3. Tissue APEX1 Measurement by Immunohistochemical Staining Analysis

Immunohistochemical staining of the FFPE tissue section was conducted as previously described [10] using a primary rabbit polyclonal antibody against human APEX1 (product # MR-PAAPE, diluted 1:50, incubation at 31 °C for 32 min; MediRedox, Daejeon, South Korea). We analyzed the nuclear and cytoplasmic APEX1 expression separately. Immunohistochemical staining was scored using the method described by Allred et al. [12]. Specifically, semiquantitative scoring from 0 to 8 was used to evaluate the cytoplasmic and nuclear expression of stained cancer cells or nontumor hepatocytes, bile duct epithelial cells, and proximal convoluted tubular epithelial cells. The results were examined separately and scored by Kim, K-H, and Yeo, M-K, who were blinded to patients’ clinicopathological details. Discrepancies in the scores were discussed to obtain a consensus.

### 2.4. APEX1 mRNA Measurement by Reverse Transcriptase Digital Droplet PCR (RT-ddPCR)

Twenty paired frozen samples of HCC and nontumor liver tissue from 20 patients with HCC were used. All nontumor tissue samples were at least 3.0 cm from the tumor mass. The tissues were dissociated with a gentle MACS dissociator with a human tumor dissociation kit (Miltenyi Biotec, Bergisch Gladbach, Germany). To remove as many nonparenchymal cells as possible, the supernatant was removed after centrifugation, and a precipitated cell suspension was used. The following primers were developed: the APEX1 forward primer sequence 5′-GGC-AAG-CTT-GAG-TCA-GGACC-3′ and the reverse primer sequence 5′-TTT-ACC-GCG-TTG-CTC-GC-3′ (input PCR template: NM_001244249.1). The QX200™ Droplet Digital™ PCR System (Bio-Rad, Pleasanton, CA, USA) was used for RT-ddPCR analysis of APEX1. RT-ddPCR was conducted as previously described [13].

### 2.5. Statistics

The relationships between APEX1 expression and the clinicopathological parameters were evaluated using the Mann–Whitney U test. Differences in APEX1 mRNA expression between the paired HCC tissue and nontumor liver tissue sections were assessed using the Wilcoxon signed-rank test. The Cox proportional hazards model was applied for univariate and multivariate survival analyses. Differences in the s-APEX1 level between the control, HCC, CC, ccRCC, and HBV DNA (+) groups were assessed using the Kruskal–Wallis test. Receiver operating characteristic (ROC) curve analysis was applied to evaluate the utility of s-APEX1 levels in determining HCC, CC, or ccRCC. Optimal criteria for sensitivity and specificity were indicated by Youden’s index using MedCalc version 19.0.5 (MEDCALC, Ostend, Belgium). Associations between protein variables were assessed using the Spearman correlations. The statistical significance was set to *p* < 0.05 (SPSS v.24; SPSS, Inc., Chicago, IL, USA).

## 3. Results

### 3.1. Association between APEX1 Expression and Clinicopathological Characteristics

APEX1 expression showed both nuclear and cytoplasmic localization in the HCC, CC, and ccRCC cells; however, nuclear expression was generally predominant (Figure 1). APEX1 expression was also shown in nontumor parenchymal and mesenchymal cells including hepatocytes, bile duct epithelial cells, renal tubular cells, stromal fibroblasts, vascular endothelial, smooth muscle cells, and inflammatory cells with nuclear, cytoplasmic, or both nuclear and cytoplasmic staining patterns. There was a significant positive correlation between nuclear or cytoplasmic APEX1 expression and histologic nuclear grade in the ccRCC cases (Table 2). There were significant positive relationships with nuclear APEX1 expression and old age (over 65 years at the time of the operation) in the HCC group (*p* = 0.018) and a positive correlation between nuclear APEX1 expression and HBV infection in the CC group (*p* = 0.018) (Table 3 and Table 4). However, cytoplasmic expressions of APEX1 were significantly higher in HCC, CC, and ccRCC cells than in the matched nontumor hepatocytes, bile duct epithelial cells, and proximal convoluted tubular epithelial cells (*p* < 0.001, each) (Figure 2).

### 3.2. Cytoplasmic APEX1 May Predict Poor Prognosis with Relapse in HCC and Intrahepatic CC Cases

Cytoplasmic APEX1 expression in HCC cells and CC cells was associated with a shorter DFS period in the univariate analysis (*p* = 0.029 and *p* = 0.024) (Appendix A). Multivariate analysis with other clinicopathological values including age, sex, HBV infection, cirrhosis, histologic grading, and pathologic TNM staging showed that cytoplasmic APEX1 expression in HCC and CC cells is also a predictor of a shorter DFS period (*p* = 0.009 and *p* = 0.011, respectively) (Table 5 and Table 6). However, nuclear or cytoplasmic APEX1 expression in survival analyses of ccRCC patients did not reach statistical significance (Appendix A).

### 3.3. APEX1 mRNA Expression Levels Were Higher in HCC Tissues than in Non-Neoplastic Liver Tissues

RT-ddPCR analysis of APEX1 mRNA in 20 matched pairs of HCC tissue and nontumor liver tissue showed higher copy numbers of APEX1 in HCC tissue than in matched non-neoplastic tissue (*p* = 0.048, Wilcoxon signed-rank test) (Appendix A). The average number of copies of APEX1 in the 50 ng cDNA template in the HCC tissue/non-neoplastic hepatic tissue was 10167.76/8.64.

### 3.4. Serum APEX1 Expression Significantly Increased in Patients with HCC, CC, and ccRCC 

The s-APEX1 level was measured by ELISA and is expressed as the median value of the s-APEX1 concentration in Figure 3. The s-APEX1 concentrations of HCC, CC, and ccRCC were between the levels of the healthy control group and the patients with HBV DNA (+). The s-APEX1 level was correlated with aspartate aminotransferase (AST), alanine aminotransferase (ALT), Gamma-glutamyl transferase (GGT), C-reactive protein (CRP), and HBV DNA (+) (Appendix A). As the s-APEX1 level was significantly higher in the HCC, CC, and ccRCC groups than in the healthy control group when the HBV DNA (+) group was excluded, we performed ROC analysis to identify the appropriateness of s-APEX1 as a diagnostic marker for the three cancers (Figure 4). S-APEX1 was undetectable (<0.00001 ng/mL) in 11/39 (28.2%) of the healthy controls, in 1/59 (1.7%) of HCC cases, in 10/46 (21.7%) of CC cases, in 1/40 (2.5%) of ccRCC cases, and in 0/32 (0.0%) of HBV DNA (+) cases.

The criteria to distinguish the HCC, CC, and ccRCC groups from the healthy control group were >0.1534 ng/mL (sensitivity: 81.4; specificity: 94.9), >0.202 ng/mL (sensitivity: 73.9; specificity: 97.4), and >0.202 ng/mL (sensitivity: 82.5; specificity: 97.4), respectively (Figure 4A–C). The criteria to distinguish the HBV DNA (+) group from the HCC, CC, and ccRCC groups were >0.5517 ng/mL (sensitivity: 93.7; specificity: 89.8), >1.1376 ng/mL (sensitivity: 75.0; specificity: 76.1), and >1.5859 ng/mL (sensitivity: 62.5; specificity: 100.0), respectively (Figure 4D–F).

## 4. Discussion

In this study, we evaluated the subcellular and extracellular expressions of APEX1 in ccRCC, HCC, and CC. We demonstrated that the s-APEX1 level was significantly higher in patients with the three cancers than in the healthy control group, while the serum level of individuals in the HBV DNA (+) group was higher than those with three cancers. In addition, increased cytoplasmic expression of APEX1 in HCC and intrahepatic CC cells was found to be associated with a shorter DFS period.

The s-APEX1 level was markedly increased in patients with HBV DNA (+), primary ccRCC, HCC, or CC, and the s-APEX1 levels exhibited positive correlations with AST, ALT, GGT, CRP, and HBV DNA (+). These findings suggest active extracellular secretion of APEX1 from cancer cells or activated stromal or inflammatory cells. Little is known about the mechanism or influence of extracellular secretion of APEX1 in the three cancers and under inflammatory conditions. In agreement with our study, a recent study showed markedly higher s-APEX1 levels in HCC patients in comparison to non-tumor cirrhosis and healthy control groups [6]. Moreover, s-APEX1 levels were found to be significantly higher in individuals with HCC-related viral infection (either viral hepatitis B or C infection) compared to those with other non-viral etiologies, but the highest s-APEX1 level was seen in patients with multiple etiologies including viral, metabolic, or alcohol-related conditions [6]. This demonstrates that APEX1 promotes IL-6 and IL-8 expression in JHH6 cells, suggesting that it influences the tumor microenvironment [6]. In our study, the highest s-APEX1 level was observed in the non-tumor HBV DNA (+) group. The HBV DNA (+) state is generally associated with an active immune reaction or inflammation, while liver cirrhosis or the surgical preparation period were relatively stable in terms of immune or inflammatory reactions because of end stage fibrotic liver or the stable conditions required for surgery. Our results exhibit that, in all study groups, the s-APEX1 level has a positive correlation with AST, ALT, GGT, and CRP. Considering the role of APEX1 in triggering an inflammation or immune reaction, we speculate that elevated levels of s-APEX1 may reflect an inflammation/immune reaction. Therefore, an active inflammation/immune state and the three cancers need to be separated when the s-APEX1 level is considered to be a diagnostic tumor marker of the three cancers. Our data showed that s-APEX1 has a statistically significant ability to differentiate the three cancers from the healthy control group and the HBV DNA (+) group. Therefore, we suggest that s-APEX1 expression could be a suitable diagnostic tumor marker for the three cancers.

APEX1 a variety of conditions ranging from physiologic to pathologic, acting as an essential protein in base excision repair pathways and as a transcriptional coactivator for ubiquitous or tissue-specific transcription factors on redox-dependent or redox-independent mechanisms. Some studies have suggested that the regulation of APEX1 functions is controlled by posttranslational modification and the heterogeneity of subcellular localization [1,14]. Different conditions change the APEX1 subcellular localization, for example, to the nucleus, mitochondria, or endoplasmic reticulum. APEX1 subcellular localization is prominently nuclear, whereas cytoplasmic APEX1 expression has also been shown in highly metabolic or proliferative non-neoplastic cells [15,16]. APEX1 modulates intracellular redox status in oxidative stress [17]. A mitochondrial targeting sequence (MTS) of APEX1 resides within residues 289–318 in the C terminus, normally masked by the intact N-terminal structure [18]. The APEX1 and Mia40 interaction in mitochondria highlights the role of maintaining mitochondrial DNA stability and cell survival [19]. Photoirradiation of a lung cancer therapy induces the mitochondrial translocation of APEX1 to control mitochondrial transcription activity by mitochondrial transcription factor A as a redox regulation [20]. HCC and ccRCC can have metabolic reprogramming to cover highly proliferative state [21]. Recent studies have been reported an important mitochondrial regulatory role for redox regulation that mitochondrial NADH dehydrogenase (ubiquinone) 1 alpha subcomplex 4-like 2 (NDUFA4L2) were overexpressed in ccRCC and HCC cells. The NDUFA4L2 overexpression by glycolysis in solid cancer cells is suggestive of one of metabolic adaptations in hypoxic state to reduce mitochondrial oxidative phosphorylation and control increased reactive oxygen species production [22,23].

In both colonic adenoma and adenocarcinoma, APEX1 expression has been shown to be pronounced in the cytoplasm, in contrast to the predominantly nuclear expression in normal colonic epithelial cells [24]. Although the molecular basis of changing the subcellular localization is not clear, the cytoplasmic distribution of APEX1 has been reported to be associated with aggressiveness in some cancers including lung, ovarian, thyroid, breast, and liver cancers [25,26,27]. In HEK293 cells derived from human embryonal kidneys, cytoplasmic APEX1 protein seems to be secreted extracellularly in response to treatment with Trichostatic A, an inhibitor of histone deacetylases [28]. Therefore, the extracellular APEX1 level could be associated with special functions. According to the TCGA database, the correlation between mRNA expression level and ccRCC patient survival rate was lower in the low-expression ccRCC patients than in the high-expression patients (https://www.proteinatlas.org/ENSG00000100823-APEX1/pathology/tissue/renal+cancer/KIRC). In our results, but, nuclear or cytoplasmic APEX1 protein expression was not statistically significant in survival analysis of ccRCC patients although there was a significant positive correlation between nuclear or cytoplasmic APEX1 expression and histologic nuclear grade in the ccRCC cases. Differences in patient survival outcome between the APEX1 mRNA and protein expression may be due to differences in APEX1 protein expression in ccRCC cells and total mRNA expression of ccRCC cells and various nontumor cells in tumor tissue sections.

Our evaluation of APEX1 expression and clinicopathological data revealed a correlation between cytoplasmic APEX1 expression in tumor cells and shorter DFS times in patients with HCC or intrahepatic CC. Cytoplasmic expression levels of APEX1 in HCC, CC, and ccRCC cells were higher than those in the matched non-neoplastic hepatocytes, bile duct epithelial cells, and proximal convoluted tubular epithelial cells. The s-APEX1 levels of the HCC, CC, and ccRCC groups were between the levels of the healthy control group and the HBV DNA (+) group and showed sufficient diagnostic accuracy to distinguish between the healthy control and the HBV DNA (+) group. 

## 5. Conclusions

The study aimed to investigate the clinicopathological significance of APEX1 expression in human HCC, CC, and ccRCC. Our study showed that s-APEX1 levels in the HCC, CC, and ccRCC groups were statistically sufficient diagnostic accuracy to distinguish the caner groups from both the healthy control group and the HBV DNA (+) group. Cytoplasmic APEX1 levels of HCC and CC cells was associated with a shorter DFS period. Despite limiting the molecular mechanism of differential expression, heterologous subcellular distribution and extracellular secretion of APEX1 might reflect its manifold activity in ccRCC, HCC, and CC. We suggest that subcellular or extracellular APEX1 expression in ccRCC, HCC, or CC could be used as a potential diagnostic or prognostic marker for tumor progression screening tests.

## Figures and Tables

**Figure 1 jcm-08-01151-f001:**
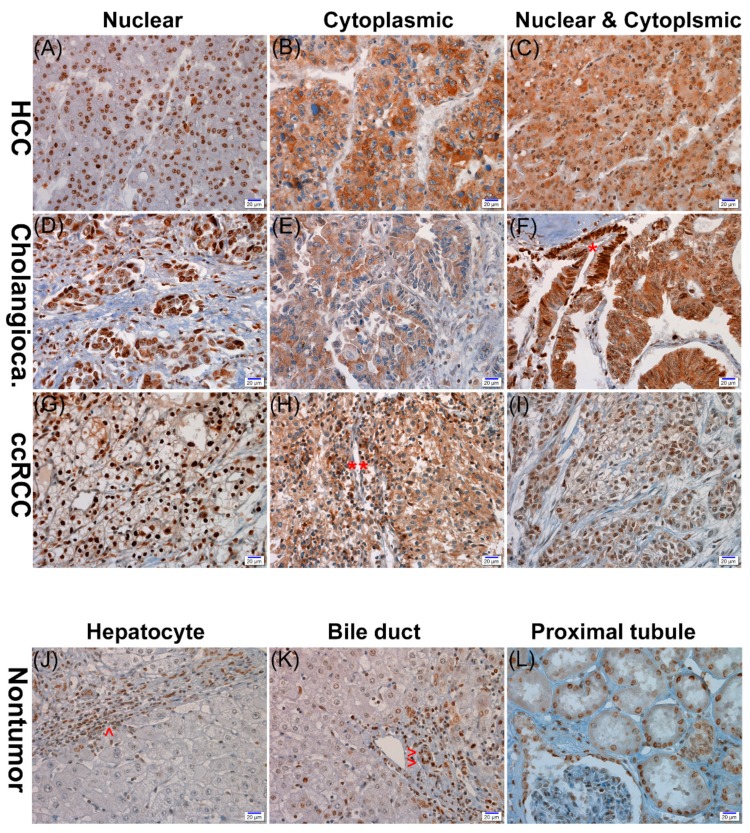
Representative photographs of APEX1 immunohistochemical staining in hepatocellular carcinoma (HCC) (**A**–**C**), cholangiocarcinoma (Cholangioca) (**D**–**F**), and clear cell renal cell carcinoma (ccRCC) (**G**–**I**). The three cancers show nuclear, cytoplasmic, or nuclear and cytoplasmic expression of APEX1. The nontumor proliferative biliary epithelium (*) expresses nuclear staining, while the cholangiocarcinoma cells show both nuclear and cytoplasmic expression (**F**). Inflammatory cells (**) exhibit cytoplasmic staining (**H**). No APEX1 expression in nontumor hepatocytes in contrast to the high nuclear expression in inflammatory cells (^) (**J**). Positive nuclear expression in nontumor bile duct epithelial cells (^^) (**K**). Positive nuclear expression in proximal convoluted tubular epithelial cells (**L**) (scale bar = 20 μm).

**Figure 2 jcm-08-01151-f002:**
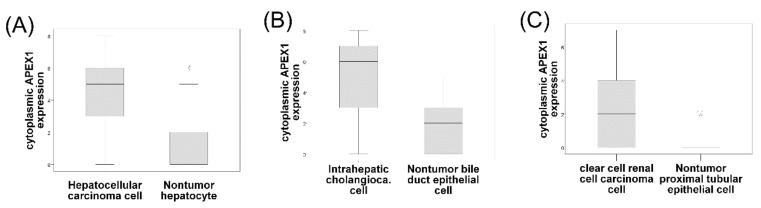
Cytoplasmic expression of the APEX1 protein in 131 paired cases of hepatocellular carcinoma cells and the matched nontumor hepatocytes (**A**), 32 paired cases of intrahepatic cholangiocarcinoma cells and the matched nontumor bile duct epithelial cells (**B**), and 106 paired cases of clear cell renal cell carcinoma cells and the matched nontumor proximal convoluted tubular epithelial cells (**C**), using immunohistochemical staining for formalin-fixed, paraffin-embedded tissue (Wilcoxon signed-rank test: *p* < 0.001, each). The line in the middle of the boxes is the median. The box length indicates the interquartile range. The ends of the whiskers represent maximum and minimum values.

**Figure 3 jcm-08-01151-f003:**
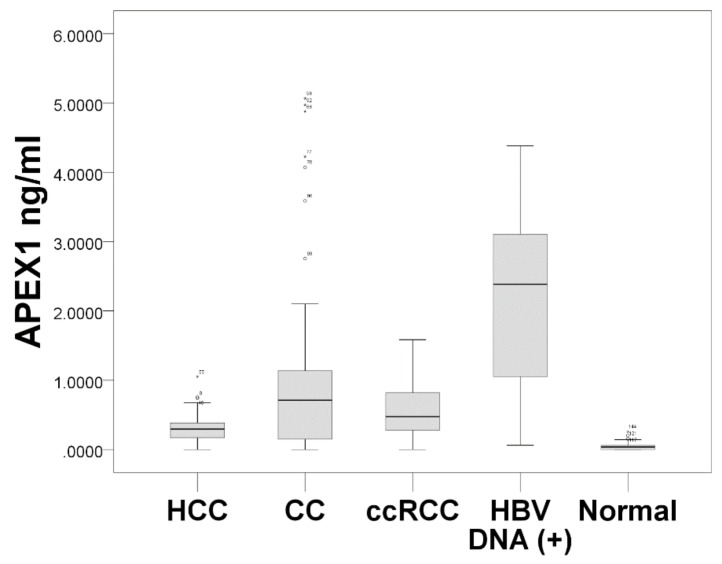
Comparison of the s-APEX1 levels among hepatocellular carcinoma (HCC), cholangiocarcinoma (CC), clear cell renal cell carcinoma (ccRCC), HBV DNA (+) (HBV DNA ≥ 58 copies/mL), and heathy control groups (Kruskal–Wallis test: *p* < 0.001). The dark line in the middle of the boxes is the median. The box length indicates the interquartile range. The ends of the whiskers represent maximum and minimum values.

**Figure 4 jcm-08-01151-f004:**
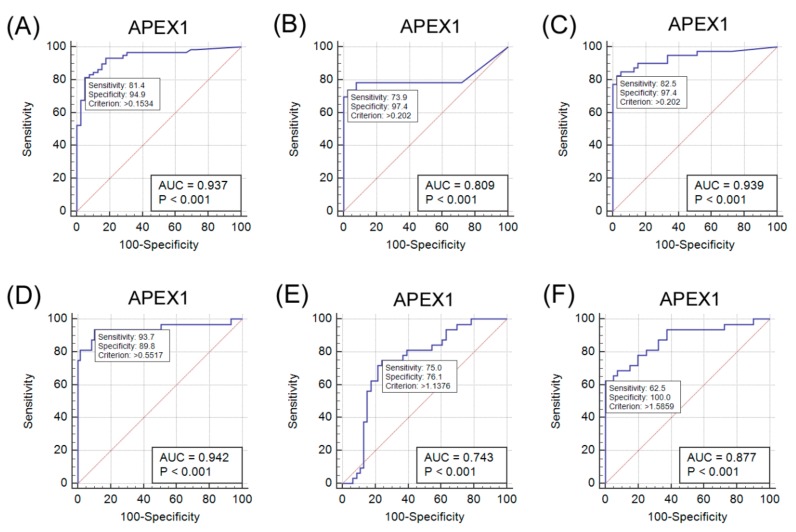
Analysis of the diagnostic value of serum APEX1 for hepatocellular carcinoma (HCC), cholangiocarcinoma (CC), and clear cell renal cell carcinoma (ccRCC) by receiver operating characteristic (ROC) curves. (**A**–**C**) ROC curves for the serum APEX1 levels to separate the HCC, CC, or ccRCC group from the healthy control. (**D**–**F**) ROC curves for the serum APE1/Red-1 levels to separate HBV DNA (+) from the HCC, CC, or ccRCC group. (**A**) ROC curve: HCC (+) vs. healthy control group (−), 95% confidence interval: 0.869–0.976. (**B**) ROC curve: CC (+) vs. healthy control group (–), 95% confidence interval: 0.709–0.886. (**C**) ROC curve: ccRCC (+) vs. healthy control group (–), 95% confidence interval: 0.862–0.981. (**D**) ROC curve: HBV DNA (+) (+) vs. HCC (–), 95% confidence interval: 0.873–0.980. (**E**) ROC curve: HBV DNA (+) (+) vs. CC (–), 95% confidence interval: 0.631–0.835. (**F**) ROC curve: HBV DNA (+) (+) vs. ccRCC (–), 95% confidence interval: 0.778–0.942.

**Table 1 jcm-08-01151-t001:** Summary of the materials from the 505 study individuals.

Case	No. Individuals
FFPE tissue specimen group	**269**
Hepatocellular carcinoma	131
HCC with HBV	106
HCC without HBV	25
Intrahepatic cholangiocarcinoma	32
Clear cell renal cell carcinoma	106
Frozen serum specimen group	**216**
Healthy control	39
Hepatitis B viral DNA (+) *	32
Hepatocellular carcinoma	59
Cholangiocarcinoma	46
Intrahepatic	13
hilar	5
distal	28
Clear cell renal cell carcinoma	40
Frozen tissue specimen group	**20**
Hepatocellular carcinoma	20

FFPE, Formalin-fixed, Paraffin-embedded; HCC: hepatocellular carcinoma; HBV: Hepatitis B viral surface antigen (+) and Hepatitis B viral surface antibody (−); * non-tumor HBV DNA (+) group: Hepatitis B viral DNA ≥ 58 copies/mL.

**Table 2 jcm-08-01151-t002:** Summary of the clinicopathologic characteristics of patients with primary clear cell renal cell carcinoma (*n* = 106).

Characteristics	No.	Nuclear APEX1		Cytoplasmic APEX1	
		Median (IQR)	*p* *	Median (IQR)	*p* *
Age at operation			0.778		0.778
≤65	54	7 (7–7)		2 (0–4)	
>65	52	7 (6.25–7)		2 (0–3.75)	
Sex			0.885		0.383
Male	75	7 (7–7)		2 (0–3)	
Female	31	7 (7–7)		2 (0–5)	
Histologic nuclear grade			0.016		<0.001
I–II	72	7 (7–7)		0 (0–2)	
III–IV	34	7 (5.75–7)		4.5 (2–6)	
Pathologic T stage			0.189		0.512
I	44	7 (7–7.75)		2 (0–4.75)	
II–IV	62	7 (6.75–7)		2 (0–3)	

* Mann-Whitney U test; IQR, interquartile range; nuclear APEX1, nuclear expression of APEX1 protein in cancer cells; and cytoplasmic APEX1, cytoplasmic expression of APEX1 protein in cancer cells.

**Table 3 jcm-08-01151-t003:** Summary of the clinicopathologic characteristics of patients with hepatocellular carcinoma (*n* = 131).

Characteristics	No.	Nuclear APEX1		Cytoplasmic APEX1	
		Median (IQR)	*p*	Median (IQR)	*p*
Age at operation			0.018 *		0.396 *
≤65	109	7 (6–7.50)		5 (3–6)	
>65	22	8 (6.75–8)		4 (2–6.25)	
Sex			0.121 *		0.355 *
Male	102	7 (6–8)		5 (3–6)	
Female	29	7 (5.5–7)		5 (4–6)	
HBV infection			0.421 *		0.747 *
No	25	7 (7–8)		5 (3.5–6)	
Yes	106	7 (6–8)		5 3–6)	
Cirrhosis			0.355 *		0.534 *
No	33	7 (5.5–8)		5 (3–6)	
Yes	98	7 (6–8)		5 (3–6)	
Tumor size (the greatest diameter)			0.174 *		0.096 *
≤5.0 cm	110	7 (6–8)		5 (3.75–6)	
>5.0 cm	21	7 (6.5–8)		4 (3–5.5)	
Lymphovascular invasion			0.559 *		0.569 *
No	31	7 (6–8)		5 (3–6)	
Yes	100	7 (6–8)		5 (3–6)	
Histologic grade			0.947 *		0.449 *
1–2 (well and moderate)	110	7 (6–8)		5 (3–6)	
3–4 (poorly and undiff.)	21	7 (5–8)		5 (3.5–7)	
Pathologic stage			0.870 **		0.663 **
I	45	7 (6–7.5)		5 (4–6)	
II	78	7 (6–8)		5 (3–6)	
III	5	7 (6.5–7.5)		6 (3.5–6.5)	
IV	3	7 (NA)		5 (NA)	

* Mann–Whitney U test; ** Kruskal–Wallis test; IQR, interquartile range; HBV infection, Hepatitis B viral surface antigen (+) and Hepatitis B viral surface antibody (−); tumor size, the greatest diameter; undiff., undifferentiated; nuclear APEX1, nuclear expression of APEX1 protein in cancer cells; cytoplasmic APEX1, cytoplasmic expression of APEX1 protein in cancer cells: and NA, not applicable.

**Table 4 jcm-08-01151-t004:** Summary of the clinicopathologic characteristics of patients with primary intrahepatic cholangiocarcinoma (*n* = 32).

Characteristics	No.	Nuclear APEX1		Cytoplasmic APEX1	
		Median (IQR)	*p*	Median (IQR)	*p*
Age at operation			0.785 *		0.696 *
≤65	21	7 (3–7)		6 (3–7)	
>65	11	6 (6–7)		6 (3–7)	
Sex			0.229 *		0.458 *
Male	23	7 (5–7)		6 3–7)	
Female	9	6 (2.5–7)		5 (2–6.5)	
HBV infection			0.018 *		0.480 *
No	24	6 (3–7)		6 (3.25–7)	
Yes	8	7 (6.25–8)		4 (3–6.5)	
Cirrhosis			0.092 *		0.207 *
No	25	7 (3–7)		6 (3.5–7)	
Yes	7	4 (6–8)		3 (3–5)	
Histologic grade			0.285 *		0.308 *
1–2 (well & moderate)	26	7 (5.75–7)		6 (3–7)	
3–4 (poorly & undiff.)	6	3.5 (0–7)		5 (0–6.25)	
Pathologic stage			0.355 **		0.099 **
I	3	7 (NA)		6 (NA)	
II	27	7 (5–7)		6 (3–7)	
III	2	NA		NA	
IV	0	NA		NA	

* Mann–Whitney U test; ** Kruskal–Wallis test; IQR, interquartile range; HBV infection, Hepatitis B viral surface antigen (+) and Hepatitis B viral surface antibody (−); undiff., undifferentiated; nuclear APEX1, nuclear expression of APEX1 protein in cancer cells; and cytoplasmic APEX1, cytoplasmic expression of APEX1 protein in cancer cells; and NA, not applicable.

**Table 5 jcm-08-01151-t005:** Multivariate analysis of overall survival and disease-free survival in 131 patients with primary hepatocellular carcinoma (HCC).

Prognostic Factor	Overall Survival		Disease-Free Survival	
	HR (95% CI)	*p **	HR (95% CI)	*p **
Cytoplasmic APEX1	1.106 (0.936–1.307)	0.237	1.153 (1.036–1.284)	0.009
Age at operation		0.161		0.908
≤65	1 (reference)		1 (reference)	
>65	0.401 (0.112–1.437)		0.963 (0.510–1.819)	
Sex		0.374		0.826
Male	1 (reference)		1 (reference)	
Female	0.629 (0.226–1.749)		0.934 (0.509–1.714)	
HBV infection		0.640		0.564
No	1 (reference)		1 (reference)	
Yes	1.259 (0.479–3.308)		1.202 (0.643–2.246)	
Cirrhosis		0.757		0.092
No	1 (reference)		1 (reference)	
Yes	0.881 (0.396–1.963)		1.645 (0.923–2.931)	
Tumor size (the greatest diameter)		0.004		0.022
≤5.0 cm	1 (reference)		1 (reference)	
>5.0 cm	4.400 (1.606–12.054)		2.219 (1.122–4.389)	
Lymphovascular invasion		0.121		0.419
No	1 (reference)		1 (reference)	
Yes	0.415 (0.137–1.262)		1.382 (0.631–3.028)	
Histologic grade		0.004		0.019
1–2 (well and moderate)	1 (reference)		1 (reference)	
3–4 (poorly and undiff.)	3.326 (1.455–7.599)		1.970 (1.116–3.479)	
Pathologic stage		0.401		0.234
I	1 (reference)		1 (reference)	
II–IV	1.525 (0.570–4.081)		1.464 (0.782–2.743)	

* multivariate Cox regression analysis; HR, hazard ratio; CI, confidence interval; HBV infection, Hepatitis B viral surface antigen (+) and Hepatitis B viral surface antibody (−); and cytoplasmic APEX1, cytoplasmic expression of APEX1 protein in cancer cells.

**Table 6 jcm-08-01151-t006:** Multivariate analysis of overall survival and disease-free survival in 32 patients with primary intrahepatic cholangiocarcinoma.

Prognostic Factor	Overall Survival		Disease-Free Survival	
	HR (95% CI)	*p **	HR (95% CI)	*p **
Cytoplasmic APEX1	1.563 (0.886–2.757)	0.123	1.407 (1.081–1.831)	0.011
Age at operation		0.122		0.343
≤65	1 (reference)		1 (reference)	
>65	0.181 (0.021–1.578)		0.596 (0.205–1.736)	
Sex		0.525		0.802
Male	1 (reference)		1 (reference)	
Female	0.585 (0.112–3.049)		1.150 (0.385–3.438)	
HBV infection		0.517		0.182
No	1 (reference)		1 (reference)	
Yes	0.469 (0.048–4.627)		0.224 (0.025–2.017)	
Cirrhosis		0.403		0.133
No	1 (reference)		1 (reference)	
Yes	3.598 (0.179–72.492)		5.979 (0.580–61.672)	
Histologic grade		0.485		0.831
1–2 (well and moderate)	1 (reference)		1 (reference)	
3–4 (poorly and undiff.)	0.459 (0.052–4.073)		1.145 (0.331–3.966)	
Pathologic stage		0.787		0.718
I	1 (reference)		1 (reference)	
II–IV	0.734 (0.078–6.943)		1.356 (0.260–7.089)	

* multivariate Cox regression analysis; HR, hazard ratio; CI, confidence interval; HBV infection, Hepatitis B viral surface antigen (+) and Hepatitis B viral surface antibody (−); and Cytoplasmic APEX1, cytoplasmic expression of APEX1 protein in cancer cells.

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
