# Peer review of "APEX1 Expression as a Potential Diagnostic Biomarker of Clear Cell Renal Cell Carcinoma and Hepatobiliary Carcinomas"

_jcm, 2019, doi:10.3390/jcm8081151_

Round 1

Reviewer 1 Report

In this manuscript entitled "APE1/Ref-1 expression as a potential diagnostic biomarker of clear cell renal cell carcinoma and hepatobiliary carcinomas”, Kim et. al. examined APE1/Ref-1 expression in clear cell renal cell carcinoma and hepatobiliary carcinomas. Even though the general elevation of APE1/Ref-1 in cancers and its potential utility as biomarkers are not novel ideas, the authors did careful and appropriate analyses and drew meaningful conclusions.

Author Response

We attached univariate and multivariate analyses for ccRCC in supplementary data and revised Figure 1 and discussion section. The revised content is highlighted.

Reviewer 2 Report

In this study was evaluated the tissue expression and serum levels of APE1/REF1 in three different tumors.

I have some comments:

- The official symbol should be used: namely APEX1

- A box and whisker plot for tissue expression of APEx1 in ccRCC and CC is missing. 

- I have evaluated the prognostic role of APEX1 expression in TGCA - KIRC database and I found that ccRCC patients with low expression had reduced survival compared to cases with higher expression. The authors should discuss their findings in comparison with TGCA data

- univariate and multivariate analyses for ccRCC are missing

-  ccRCC and HCC are characterized by reprogramming of energetic metabolism. In particular the metabolic flux through glycolysis is partitioned (PMID: 30983433), and mitochondrial bioenergetics and OxPhox are impaired (PMID: 30538212  and PMID:26819450) resulting in ROS production. These findings should be discussed and integrated with the study results. 

Author Response

In the revised manuscript, we discussed and described everything you commented on.  The revised content is highlighted. 

- The official symbol should be used: namely APEX1.

  : We have revised APE1/Ref-1 to APEX1.

- A box and whisker plot for tissue expression of APEx1 in ccRCC and CC is missing. 

  : We inserted it in figure 2.

- I have evaluated the prognostic role of APEX1 expression in TGCA - KIRC database and I found that ccRCC patients with low expression had reduced survival compared to cases with higher expression. The authors should discuss their findings in comparison with TGCA data

 : We commented it in discussion section. 

. According to the TCGA database, the correlation between mRNA expression level and ccRCC patient survival rate was lower in the low-expression ccRCC patients than in the high-expression patients (https://www.proteinatlas.org/ENSG00000100823-APEX1/pathology/tissue/renal+cancer/KIRC). In our results, but, nuclear or cytoplasmic APEX1 protein expression was not statistically significant in survival analysis of ccRCC patients although there was a significant positive correlation between nuclear or cytoplasmic APEX1 expression and histologic nuclear grade in the ccRCC cases. Differences in patient survival outcome between the APEX1 mRNA and protein expression may be due to differences in APEX1 protein expression in ccRCC cells and total mRNA expression of ccRCC cells and various nontumor cells in tumor tissue sections.

- univariate and multivariate analyses for ccRCC are missing

 : We inserted it in supplementary table 3 and 4.

-  ccRCC and HCC are characterized by reprogramming of energetic metabolism. In particular the metabolic flux through glycolysis is partitioned (PMID: 30983433), and mitochondrial bioenergetics and OxPhox are impaired (PMID: 30538212  and PMID:26819450) resulting in ROS production. These findings should be discussed and integrated with the study results. 

 : We commented it in discussion section. 

APEX1 modulates intracellular redox status in oxidative stress [17]. A mitochondrial targeting sequence (MTS) of APEX1 resides within residues 289-318 in the C terminus, normally masked by the intact N-terminal structure [18]. The APEX1 and Mia40 interaction in mitochondria highlights the role of maintaining mitochondrial DNA stability and cell survival [19]. Photoirradiation of an lung cancer therapy induces the mitochondrial translocation of APEX1 to control mitochondrial transcription activity by mitochondrial transcription factor A as a redox regulation [20]. HCC and ccRCC can have metabolic reprogramming to cover highly proliferative state [21]. The recent studies have been reported an important mitochondrial regulatory role for redox regulation that mitochondrial NADH dehydrogenase (ubiquinone) 1 alpha subcomplex 4‐like 2 (NDUFA4L2) were overexpressed in ccRCC and HCC cells. The NDUFA4L2 overexpression by glycolysis in solid cancer cells is suggestive of one of metabolic adaptations in hypoxic state to reduce mitochondrial oxidative phosphorylation and control increased reactive oxygen species production [22-23].

Reviewer 3 Report

The authors detected the expression serum levels of APE1/Ref-1 in clinical samples and found the correlation of APE1/Ref-1 and carcinoma. It's interesting. However, the  APE1/Ref-1 immunohistochemical stainings of para-tumor tissues are missing in Figure 1. It's hard to tell the expression of APE1/Ref-1.

Author Response

We revised figure 1 to show the APE1/Ref-1 immunohistochemical stainings of non-tumor tissues.    

Round 2

Reviewer 2 Report

No comments

Reviewer 3 Report

Thank you for answering the question!